# Association between Internalized Stigma and Depression among People Living with HIV in Thailand

**DOI:** 10.3390/ijerph19084471

**Published:** 2022-04-08

**Authors:** Darawan Thapinta, Kriengkrai Srithanaviboonchai, Penpaktr Uthis, Sunisa Suktrakul, Rangsima Wiwatwongnawa, Arunrat Tangmunkongvorakul, Saranya Wannachaiyakul, Patumrat Sripan

**Affiliations:** 1Faculty of Nursing, Chiang Mai University, Chiang Mai 50200, Thailand; darawan1955@gmail.com (D.T.); saranya.wan@cmu.ac.th (S.W.); 2Research Institute for Health Sciences, Chiang Mai University, Chiang Mai 50200, Thailand; arunrat.tang@cmu.ac.th (A.T.); pspatumrat3@gmail.com (P.S.); 3Faculty of Medicine, Chiang Mai University, Chiang Mai 50200, Thailand; 4Faculty of Nursing, Chulalongkorn University, Bangkok 10330, Thailand; penpaktr_uthis@yahoo.com (P.U.); auyyoo@yahoo.com (S.S.); 5Faculty of Social Sciences, Chiang Mai University, Chiang Mai 50200, Thailand; rwiwatwongwana@gmail.com

**Keywords:** internalized stigma, depression, people living with HIV, Thailand

## Abstract

Internalized stigma and depression are among the most common mental health problems in people living with HIV (PLHIV). This study aimed to examine the association between depression and overall internalized stigma, as well as different aspects of internalized stigma in PLHIV. The study included 400 PLHIV receiving care in Bangkok and Chiang Mai, Thailand. Data were analyzed using descriptive statistics, Mann-Whitney U test, and Spearman correlation coefficients. The results indicated the PLHIV with mild depression had lower median scores for the social relationship internalized stigma subscale than PLHIV with major depressive disorder (*p* = 0.009). Total HIV internalized stigma scores were significantly correlated with PHQ-9 scores in the mild depression group (r = 0.327, *p* = 0.004). Depression and internalized stigma were prevalent and associated, especially in the area of social relationships. Health personnel should be aware of possible depression in PLHIV who have internalized stigma. Intervention to promote understanding and social support for PLHIV is warranted.

## 1. Introduction

HIV/AIDS remains a major public health concern around the world. According to the Joint United Nations Program on HIV/AIDS (UNAIDS), there were approximately 1.5 million new HIV infections, 680,000 deaths, and 37.7 million people living with HIV (PLHIV) in 2020 [1]. UNAIDS set global targets to reach the 90-90-90 targets by 2020 given widespread access to highly effective antiretroviral medications. A recent assessment found that the targets were missed by a considerable margin, and HIV stigma was mentioned as one of the key obstacles [2,3,4].

Stigma is a Greek term denoting a mark that was burned or cut into the flesh of an unsavory character, traitor, criminal, or slave in ancient times [5]. In our modern scientific world, Erving Goffman’s definition of stigma is largely recognized. Stigma, according to Goffman, is an unfavorable social label that alters people’s perceptions of themselves and bars them from full social acceptance [6]. HIV stigma refers to society’s negative views and perceptions about people living with HIV. Lack of knowledge about HIV which led to fear of contracting HIV, negative social perceptions about HIV and PLHIV, and moral judgement toward PLHIV were cited as the drivers of HIV stigma [7].

Graham Scambler categorized stigma as ‘external’ and ‘internal’ [8]. In the HIV context, ‘external stigma’ refers to unfair treatment experienced by PLHIV from others, while ‘internal stigma’ (internalized stigma) refers to socially constructed views and negative stereotypes about HIV/AIDS which PLHIV have incorporated into their self-concept. Internalized stigma is made up of three steps: becoming aware of the stereotype, agreeing with it, and applying it to oneself [9]. In other words, internalized stigma is the loss of a person’s self-esteem or feeling of self-worth as a result of the individual’s belief that he or she is socially unacceptable.

Internalized stigma, which is the subject of this study, can be just as harmful as external stigma. Unlike external stigma, which requires more extensive intervention, internalized stigma can be self-managed. Internalized stigma has been linked to lower levels of HIV testing [10], late linkage to care and treatment [11], decreased treatment adherence [12], and lower levels of viral suppression among those on treatment [13].

Mental health problems are more common in PLHIV, with research showing that 28–62 percent of PLHIV having some form of mental illness [14] with depression the most common mental health issue among those living with HIV. A recent report on the global prevalence of depression in PLHIV compared six World Health Organization (WHO) regions and found that the South-East Asia region had the highest prevalence of depression (40%) [15]. In Thailand, one study documented that PLHIV had high depression mean scores (12.9 out of 15) [16].

Depression aggravates HIV infection in many ways; however, this can be difficult to diagnose in PLHIV because the signs and symptoms can be confounded with HIV illness symptoms. For example, weight loss, exhaustion, sleeplessness, and anorexia are all linked to both depression and HIV. As a result, depression in PLHIV is frequently undiagnosed and undertreated [17,18]. Adherence to antiretroviral medicine is also affected by depression, which can hasten the progression of HIV disease [19]. While depression among PLHIV has been extensively researched around the world, only a few studies have been conducted in Asian developing countries such as Thailand [20].

Several studies from Asia, Africa, Australia, the Caribbean, and the United States have shown a correlation between internalized stigma and depression among PLHIV [16,21,22,23,24,25,26,27,28,29,30]. From a cognitive theory perspective, depression is defined by people’s dysfunctional negative beliefs about themselves, their life experience, their future, and the world in general [31], while internalized stigma entails accepting society’s negative attitudes and feelings regarding PLHIV and applying them to oneself [32]. These similarities could explain why depression has been linked to internalized stigma.

In Thailand, only one study has explored the relationship between internalized stigma and depression among PLHIV [16]. This cross-sectional study was conducted in 2007 and found that internalized stigma was a predictor of depression after controlling for gender, age, income, and education. Because societal conditions have changed dramatically since the previous study, and a better measurement of internalized stigma among PLHIV has since been created, it is important for this topic to be revisited. The current study used a multidimensional measure of internalized HIV stigma [33] to describe levels of total internalized stigma as well as levels in internalized stigma subscales. The study also documented the prevalence of depression and investigated the connections between HIV internalized stigma and depression among Thai PLHIV.

## 2. Materials and Methods

*Design:* This study used a cross-sectional descriptive correlational design.

### 2.1. Study Participants and Setting

This study used data collected in Bangkok and in Chiang Mai, located in Northern Thailand. Four hundred participants were required to achieve a 95% confidence level and 5% precision, given an expected depression prevalence of 50% among PLHIV (Daniel, 1999). Inclusion criteria included HIV positive diagnosis and awareness of HIV positive diagnosis for one or more years, being 18 years or older, and receiving care at a government hospital. There were 400 participants in the study. Their demographic characteristics are shown in Table 1.

### 2.2. Data Collection

Data collection took place at the antiretroviral clinics of participating hospitals. Researchers visited the clinics when PLHIV had appointments. First, clinic staff gave a brief introduction about the study to the PLHIV and those who showed interest were invited to meet with trained interviewers in a private room. The interviewers gave a more detailed explanation of the study, screened for eligibility, and conducted the informed consent process. Information was collected using face-to-face interviews technique.

### 2.3. Measures

A demographic questionnaire was developed by the research team for the Thai context and included questions on age, gender, marital status, religion, educational attainment, occupation, type of health insurance, family income, and sufficiency of income.The Internalized HIV Stigma scale used in this study was developed by Sayles et, al. [28]. It was translated into Thai using Brislin’s back translation protocol. The measurement was comprised of 28 items categorized into four HIV internalized stigma subscales: (1) HIV stereotypes (12 items), (2) HIV disclosure concerns (five items), (3) Social relationship (seven items), and (4) Self-acceptance (four items). Responses for each item used a five-point categorical response scale (none of the time, a little of the time, some of the time, most of the time, or all of the time). Total mean scores were transformed linearly to a 0–100 range and described as total HIV internalized stigma score, with lower scores reflecting less perception and fewer experiences of internalized HIV stigma, and higher scores reflecting higher levels of stigma. The Cronbach’s alpha coefficient for this study was 0.92 for total HIV internalized stigma score, 0.90 for HIV stereotypes internalized stigma subscale, 0.93 for HIV disclosure internalized stigma concerns subscale, 0.89 for social relationship subscale, and 0.21 for self-acceptance subscale.Depression was assessed using the PHQ-9 (Thai version). The PHQ-9 was comprised of nine items with each item rated from “0” (not at all) to “3” (nearly every day) and total scores ranging from 0 to 27. The total depression score was categorized into five categories: 0–4 = no depression, 5–8 = mild depression, 9–14 = major depressive disorder (mild or dysthymia), 15–19 = major depressive disorder (moderate), and ≥20 = major depressive disorder (severe). The instrument had satisfactory internal consistency with Cronbach’s alpha = 0.83.

### 2.4. Statistical Analysis

Descriptive statistics were used to describe participant demographics, HIV-related internalized stigma, and depression scores. The Mann-Whitney U test was used to compare HIV-related internalized stigma scores (total and four subscales) based on depression status. Spearman correlation coefficients were used to examine the correlation between total HIV-related internalized stigma score and depression score in participants who had some form of depression, namely mild depression (PHQ-9 score 5–8) and major depression (PHQ-9 score ≥ 9). Participants who did not have depression (PHQ-9 score 0–4) were not included in this correlational analysis.

### 2.5. Ethical Considerations

This study was approved by the Human Experimentation Committee at the Research Institute for Health Sciences, Chiang Mai University (Certificate number 24/2019). All participation was voluntary and written informed consent was obtained from each participant. Participants who were found to have major depressive disorder were referred to care appropriate to coverage provided by their respective health insurance schemes.

## 3. Results

Table 1 shows detailed demographic characteristics of the participants.

The median and IQR score of total internalized stigma score for all participants was 71.0 (60.2–80.8). Subscale scores were HIV stereotypes, median and IQR score of 31.0 (25.0–37.0); HIV disclosure concerns, median and IQR score of 14.0 (10.0–18.0); social relationship, median and IQR score of 14.0 (10.0–15.0); and self-acceptance, median and IQR score of 12.0 (11.0–14.0) (Table 2).

Ninety-six participants (24.0%) could be defined as having depression. Of these, 74 (18.5%) had mild depression while 22 (5.5%) had major depressive disorder. Median scores of total HIV internalized stigma in the major depressive disorder group (78.0) was higher than the mild depression group (77.0) and the group without depression (67.0) (Table 2).

Comparisons of the total HIV internalized stigma score and the subscales using the Mann-Whitney U test showed the scores for social relationship internalized stigma subscale were significantly different (*p* < 0.01) between participants with mild depression (14.0) and those with major depressive disorder (18.0). However, no significant difference was found in the other comparisons (Table 3).

The Spearman correlation showed that total HIV internalized stigma scores were significantly correlated with PHQ-9 scores in the mild depression group (r = 0.327, *p* = 0.004). However, no correlation was found between total HIV internalized stigma scores and PHQ-9 scores of the major depressive disorder group (Table 4).

## 4. Discussion

Based on the PHQ-9, 24% of our participants were considered to be depressed, consistent with the range of 12.8–78% reported in a global systematic review [34], though our rate was lower than the 37.9% reported in another systematic review of depression in PLHIV in Ethiopia [35]. The prevalence of depression in this study was lower than the rate reported in other Thai studies. These included the prevalence studies in Thai adult PLHIV (32.2%) [36] and in Thai adolescents with vertically acquired HIV (39.3%) [37]. Differences in participant characteristics, dates of data collection, and the measurements employed could explain the variations in results.

Several research studies have indicated a link between internalized stigma and depression in PLHIV. The phenomenon was consistently found in general adults living with HIV [16,22,23,26,38], women living with HIV [21], older adults living with HIV (aged ≥ 50 years old) [24], and MSM [25]. However, these studies only examined overall internalized HIV stigma. The current study had the unique opportunity to investigate the link between different domains of internalized HIV stigma and depression, having received permission to use a scale that measured both overall HIV internalized stigma and different aspects of internalized stigma [33]. More specific results found in the study might aid healthcare professionals in developing a more targeted solution to address the issue.

In this study, an analysis was performed to determine if total internalized stigma and/or internalized stigma as represented in the four subscales differed between PLHIV with mild depression and those with major depressive disorder. Knowledge of potential differences is important, since major depressive disorder is a serious health issue that necessitates treatment. Only the median scores for the internalized stigma social relationship subscale had a significant difference, with the mild depression group scoring lower than the major depressive disorder group. This is a logical finding given the evidence that healthy social relationships are necessary for psychological well-being, particularly for PLHIV. For example, high levels of perceived social support were linked to lower levels of perceived HIV stigma [39], a precursor of internalized HIV stigma [40]. In a study of older adults living with HIV in Namibia, social support from friends had a significant negative relationship with symptoms of depression [41]. PLHIV with lower spirituality and participation in the religious community had more symptoms of depression than those with greater spirituality and religious participation [42]. Internalized stigma was also found to be influenced by how PLHIV viewed their social position. In a study among women living with HIV in the United States, internalized HIV stigma was negatively linked with greater neighborhood racial diversity, while internalized HIV stigma was positively associated with higher neighborhood household income [43].

Total internalized HIV stigma scores were positively associated with PHQ-9 scores in the mild depression group, but this was not true for the major depressive disorder group. In other words, an incrementally increased level of depression was linked to a higher likelihood of internalized stigma among our participants. The reason why this correlation was not found in the major depressive disorder group should be further examined. The small sample size of this study may explain the differing findings between the two groups.

The findings from this study highlighted the need to reduce internalized stigma, particularly among PLHIV with mild depression, in the hopes of averting a more serious condition. Interventions should promote social support among PLHIV, as perceived social support was linked to less stigma and depressed symptoms in this population [44]. To promote understanding and social support for PLHIV, community action to reduce HIV stigma is required. According to a study conducted in Thailand, community mobilization activities, which included monthly campaigns, funfairs, and information and educational materials helped to reduce HIV stigma at the community level [45]. A South African study also found that having a treatment buddy helped PLHIV participants feel less stigmatized [46]. From the PLHIV perspective, it was discovered that disclosing HIV status to one trusted family member was the first step in gaining more involvement and support from others [47]. In an African study, youth living with HIV whose friends and family members continued to socialize with them after disclosure of their HIV status had higher levels of perceived social support [48]. Another study from Africa also found that Disclosure of HIV status is associated with increased adherence to clinic visits [49]. The intervention, of course, is dependent on the individuals’ HIV disclosure status as well as the stigma experience in their specific environment.

This study’s strength was that it was the first in Thailand to examine HIV internalized stigma in PLHIV at the subscale level. This study showed that having more severe depression increased the likelihood of internalized stigma for social relationships, while having mild depression increased risk of overall internalized stigma among PLHIV. The relatively poor socio-economic status of the participants might play an important role in these findings. Some limitations included small sample size and the fact that participants were not chosen at random and represented a specific subgroup of PLHIV, specifically those receiving treatment from government programs. Additionally, the major depressive disorder group was only represented by a small number of individuals, resulting in limited statistical power when calculating correlations for the total HIV internalized stigma score. Another disadvantage is that this study was cross-sectional, preventing the assessment of temporal correlations.

Recommendations for future study include studies with a larger sample size and inclusion of participants with more diverse backgrounds. An interventional study to improve social relationships for PLHIV would be promising, as such an intervention would improve overall mental health while addressing internalized stigma.

## 5. Conclusions

Depression and internalized stigma were prevalent and associated with each other among Thai PLHIV who took part in this study, especially in the area of social relationships. Health personnel should be aware of possible depression in PLHIV who have internalized stigma. Intervention to promote understanding and social support for PLHIV is warranted.

## Figures and Tables

**Table 1 ijerph-19-04471-t001:** Participant characteristics.

Characteristics	*n* (%)
Sex	
Male	221 (55.3)
Female	179 (44.7)
Age (Median, IQR)	44 (36.0–52.0)
Marital Status	
Single	156 (39.0)
Married (living together)	152 (38.0)
Married (separated)	34 (8.5)
Divorced/Widowed	58 (13.5)
Religion	
Buddhism	379 (94.8)
Christianity	15 (3.8)
Islam	4 (1.0)
No religion	2 (0.4)
Education	
Never went to school	8 (1.8)
Primary school	120 (30.0)
Junior High School	73 (18.3)
Senior High School/ Vocational/Certificate	117 (29.3)
Bachelor’s degree or higher	82 (20.5)
Occupation	
Individually owned business	82 (20.5)
General employment	145 (36.0)
Private company employee	71 (17.8)
Government official	32 (8.0)
State enterprise official	10 (2.5)
Household business	5 (1.4)
Agriculture	24 (6.0)
No job	31 (7.8)
Type of Health Coverage	
Universal health coverage (Gold card)	233 (58.2)
Social security	132 (33.0)
Government officer	29 (7.2)
Health care for migrant workers	1 (0.3%)
No health coverage	5 (1.3%)
Household Income (Thai Baht)	
Yearly income (Median, IQR)	180,000 (106,800–300,000)
Perceived Financial Status	
Enough to save	45 (11.2)
Enough to spend	222 (55.5)
Not enough to spend	133 (33.3)
Total	400 (100.0)

**Table 2 ijerph-19-04471-t002:** Total HIV internalized stigma score and subscale scores by depression category.

Median (IQR)	No Depression (*n* = 304)	Mild Depression (*n* = 74)	Major Depressive Disorder (*n* = 22)	Total (*n* = 400)
Total HIV internalized stigma score	67.0 (59.0–77.0)	77.0 (68.5–88.0)	78.0 (69.5–94.5)	71.0 (60.2–80.8)
HIV stereotypes subscale	29.0 (24.2–36.0)	34.0 (28.0–39.2)	34.0 (28.0–38.0)	31.0 (25.0–37.0)
HIV disclosure concerns subscale	13.0 (10.0–18.0)	15.0 (13.0–19.2)	15.0 (12.8–23.3)	14.0 (10.0–18.0)
Social relationships subscale	14.0 (9.0–14.0)	14.0 (12.0–16.0)	18.0 (13.8–20.0)	14.0 (10.0–15.0)
Self-acceptance subscale	12.0 (11.0–14.0)	12.0 (11.0–15.0)	13.5 (9.8–15.3)	12.0 (11.0–14.0)

**Table 3 ijerph-19-04471-t003:** Comparison of HIV internalized stigma scores between mild depression and major depressive disorder groups.

	Mild Depression (*n* = 74)		Major Depressive Disorder (*n* = 22)		Mann-Whitney U	*p*-Value
	Median	(IQR)	Median	(IQR)		
Total HIV internalized stigma score	77.0	68.5–88.0	78.0	69.5–94.5	729.5	0.461
HIV stereotypes subscale	34.0	28.0–39.2	34.0	28.0–38.0	789.0	0.827
HIV disclosure concerns subscale	15.0	13.0–19.2	15.0	12.8–23.3	774.5	0.729
Social relationship subscale	14.0	12.0–16.0	18.0	13.8–20.0	515.0	0.009 *
Self-acceptance subscale	12.0	11.0–15.0	13.5	9.8–15.3	783.0	0.785

* *p* < 0.01.

**Table 4 ijerph-19-04471-t004:** Correlations between total HIV internalized stigma score and PHQ-9 score in the mild depression and major depressive disorder groups.

	Total HIV Internalized Stigma Score (Median and IQR)	Spearman Correlation	*p*-Value
Mild depression group (score 5–8) (*n* = 74)	77.0 (68.5–88.0)	0.327	0.004 *
Major depressive disorder group (score ≥ 9) (*n* = 22)	78.0 (69.5–94.5)	0.013	0.953

* *p* < 0.01.

## Data Availability

The data presented in this study are openly available in FigShare at http://doi.org/10.6084/m9.figshare.19333958 (accessed on 10 March 2022).

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
