# Peer review of "Association between Internalized Stigma and Depression among People Living with HIV in Thailand"

_ijerph, 2022, doi:10.3390/ijerph19084471_

Round 1
Reviewer 1 Report
Dear Authors,
I suggest broadening the conclusions.
They are too synthetic to make other reflections.
The authors support their thesis only with a small sample (n.400):Why no assumptions on gender differences were made? Why not reflect on socio-economic status of the participants from the data collected? For instance, they could provide proper explanation for the sentence "This study showed that having more severe depression increased the likelihood of internalized stigma for social relationships while having mild depression increased risk of overall internalized stigma among PLHIV".
Best regards.
Author Response
Reviewer 1:
I suggest broadening the conclusions. They are too synthetic to make other reflections.
The conclusions have been broadened per the suggestion and is now read “Depression and internalized stigma were prevalent and associated with each other among Thai PLHIV who took part in this study, especially in the area of social relationships. Health personnel should be aware of possible depression in PLHIV who have internalized stigma. Intervention to improve social relationships with others among PLHIV is warrant.”
The authors support their thesis only with a small sample (n.400):
Small sample size was added as another limitation of the study.
Why no assumptions on gender differences were made?
Thank you for raising this very important and interesting point. However, the study was designed and powered to examine the overall relationship between internalized stigma and depression. There would be too complicated, and the samples would not enough to run the sub-group analyses of genders and other demographic variables, such as age and marital status.
Why not reflect on socio-economic status of the participants from the data collected? For instance, they could provide proper explanation for the sentence "This study showed that having more severe depression increased the likelihood of internalized stigma for social relationships while having mild depression increased risk of overall internalized stigma among PLHIV".
Thank you very much for the comment. A sentence to reflect socio-economic status of the participants and further discuss the sentence has been added in the discussion session. It is read “Relatively poor socio-economic status of the participants might play an important role in this findings.”
Reviewer 2 Report
Introduction
Stigma is one of the most important sociological issues, and it seems to be the basic concept of the paper, therefore introduction might be supplemented by more sociological approaches to this social phenomenon, especially regarding HIV stigma.
The Introduction section is well prepared, although the references might be extended to a larger number of literature items, e.g. those analysing that phenomenon in other countries. It would be background to valuable comparisons.
The term ‘Internalized stigma’ should be explained more specifically in order to help potential readers to understand that phenomenon.
The description of a correlation between internalized stigma and depression has no indication what geographical region it concerns (line 64-69)
Material and Methods
In my opinion this part of paper needs some improvement.
it would be easier for the readers to understand methodological issues if the authors sort out the particular parts of this section. Study participants, data collection, measures, statistical analysis, and ethical considerations may be considered as more appropriate order.
Some information in this part needs the clarification: What was the estimated population of which the sample was 400, and where these data were obtained from? Were there any exclusion criteria? e.g. very poor health, etc.
Results
There is no need to give sociodemographic characteristic of the studied group in ‘Results’ part. It can be moved to the Material and methods section and presented in short way with reference to Table 1.
Discussion
The supplementation of the Discussion section with more references would make it more valuable.
Author Response
Reviewer 2:
Introduction
Stigma is one of the most important sociological issues, and it seems to be the basic concept of the paper, therefore introduction might be supplemented by more sociological approaches to this social phenomenon, especially regarding HIV stigma.
Societal perspective of HIV stigma and its reference has been added into the Introduction section. The sentences are read “HIV stigma refers to society's negative views and perceptions about people living with HIV. Lack of knowledge about HIV which led to fear of contracting HIV, negative social perceptions about HIV and PLHIV, and moral judgement toward PLHIV were cited as the drivers of HIV stigma (7).”
The Introduction section is well prepared, although the references might be extended to a larger number of literature items, e.g. those analysing that phenomenon in other countries. It would be background to valuable comparisons.
More references have been added to show geographical variety of published articles about association between internalized stigma and depression among PLHIV. The narrative has also adjusted to reflect this. The sentence is now read “Several studies from Asia, Africa, Australia, Caribbean, and the United States have shown a correlation between internalized stigma and depression among PLHIV [16, 21-30].”
The term ‘Internalized stigma’ should be explained more specifically in order to help potential readers to understand that phenomenon.
Additional explanation of the term “internalized stigma” has been added into the Introduction. The sentences are read “Internalized stigma is made up of three steps: becoming aware of the stereotype, agreeing with it, and applying it to oneself (8). In other words, internalized stigma is the loss of a person's self-esteem or feeling of self-worth as a result of the individual's belief that he or she is socially unacceptable.”
The description of a correlation between internalized stigma and depression has no indication what geographical region it concerns (line 64-69)
The geographical regions of the cited studies have been added. The sentence is now read “Several studies from Asia, Africa, and the United States have shown a correlation between internalized stigma and depression among PLHIV [14, 19-25].”
Material and Methods
In my opinion this part of paper needs some improvement.
it would be easier for the readers to understand methodological issues if the authors sort out the particular parts of this section. Study participants, data collection, measures, statistical analysis, and ethical considerations may be considered as more appropriate order.
Thank you very much for this valuable comment. The subtitles were re-named and the contents were re-ordered per the suggestion.
Some information in this part needs the clarification: What was the estimated population of which the sample was 400, and where these data were obtained from? Were there any exclusion criteria? e.g. very poor health, etc.
We did not know the population number, so the sample size formular for infinite population (number of population not known) was used. There was no exclusion criteria for the study. Participants were recruited at out-patient clinics so all of them were already healthy enough to participate.
Results
There is no need to give sociodemographic characteristic of the studied group in ‘Results’ part. It can be moved to the Material and methods section and presented in short way with reference to Table 1.
The sociodemographic characteristic of the participants in the Result session has been removed. Brief explanation with reference to Table 1 was added under the Study participants sub-section of the Material and methods section.
Discussion
The supplementation of the Discussion section with more references would make it more valuable.
Four references and corresponding texts have been added into the Discussion section per the suggestion.
Reviewer 3 Report
This is a well-written and well-organized manuscript. It was a pleasure to read.
The authors provide a nice overview of the literature and make a clear case for how their study fills a gap in an already highly researched area. I appreciate that they are filling a need for more research focused on PLHIV in Thailand.
I have just a few minor points:
page 2 lines 86-87: recommend changing language from "HIV-infected" as this is stigmatizing to something like "HIV positive diagnosis and knowledge of HIV positive diagnosis for one or more years"
page 2 line 97: I think this should say 5 point scale as 5 response options are listed
What was the purpose of the linear transformation of HIV stigma mean score? Is this part of the original measure instructions?
page 6, line 173: recommend changing "vertically HIV-infected" to "and in Thai adolescent with vertically acquired HIV". Language like "living with" "acquired" or "transmitted" is less stigmatizing than "infected." Additionally, as much as possible I tend to put the person or noun first, and then the descriptor. See https://www.apa.org/about/apa/equity-diversity-inclusion/language-guidelines.pdf for more information.
Author Response
Reviewer 3:
This is a well-written and well-organized manuscript. It was a pleasure to read.
The authors provide a nice overview of the literature and make a clear case for how their study fills a gap in an already highly researched area. I appreciate that they are filling a need for more research focused on PLHIV in Thailand.
I have just a few minor points:
page 2 lines 86-87: recommend changing language from "HIV-infected" as this is stigmatizing to something like "HIV positive diagnosis and knowledge of HIV positive diagnosis for one or more years"
The sentence has been revised per the suggestion and now read “Inclusion criteria included HIV positive diagnosis and aware of HIV positive diagnosis for one or more years,…”
page 2 line 97: I think this should say 5 point scale as 5 response options are listed
Thank you very much for pointing out this error. It was fixed in the revised manuscript.
What was the purpose of the linear transformation of HIV stigma mean score? Is this part of the original measure instructions?
The linear transformations were performed to be able to compare different subscales of internalized stigma scores. And yes, we followed the original measure instruction (Sayles et al, 2008).
page 6, line 173: recommend changing "vertically HIV-infected" to "and in Thai adolescent with vertically acquired HIV". Language like "living with" "acquired" or "transmitted" is less stigmatizing than "infected." Additionally, as much as possible I tend to put the person or noun first, and then the descriptor. See https://www.apa.org/about/apa/equity-diversity-inclusion/language-guidelines.pdf for more information.
The sentence has been revised per the suggestion and now read “These included the prevalence studies in Thai adult PLHIV (32.2%) [31] and in Thai adolescent with vertically acquired HIV (39.3%)”.